# Transcriptional Profiles Associated with Marek’s Disease Virus in Bursa and Spleen Lymphocytes Reveal Contrasting Immune Responses during Early Cytolytic Infection

**DOI:** 10.3390/v12030354

**Published:** 2020-03-23

**Authors:** Huan Jin, Zimeng Kong, Arslan Mehboob, Bo Jiang, Jian Xu, Yunhong Cai, Wenxiao Liu, Jiabing Hong, Yongqing Li

**Affiliations:** 1Research Center for Infectious Disease in Livestock and Poultry, Beijing Academy of Agricultural and Forestry Sciences, Beijing 100097, China; jinhuan0717@126.com (H.J.); djkanamycin@163.com (Z.K.); arslangcaas@yahoo.com (A.M.); jiangbo@cau.edu.cn (B.J.); jianxu8612@163.com (J.X.); cc19104@163.com (Y.C.); lwx232210809@163.com (W.L.); hongjb1551263132@163.com (J.H.); 2Institute of Animal Husbandry and Veterinary Medicine, Beijing Academy of Agricultural and Forestry Sciences, Beijing 100097, China; 3College of Animal Science and Technology, Jiangxi Agricultural University, Nanchang 330045, China

**Keywords:** Marek’s disease virus, lymphocytes, differentially expressed genes, cytokine, chemokine

## Abstract

Marek’s disease virus (MDV), an alpha herpes virus, causes a lymphoproliferative state in chickens known as Marek’s disease (MD), resulting in severe monetary losses to the poultry industry. Because lymphocytes of bursa of Fabricius and spleen are prime targets of MDV replication during the early cytolytic phase of infection, the immune response in bursa and spleen should be the foundation of late immunity induced by MDV. However, the mechanism of the MDV-mediated host immune response in lymphocytes in the early stage is poorly understood. The present study is primarily aimed at identifying the crucial genes and significant pathways involved in the immune response of chickens infected with MDV CVI988 and the very virulent RB1B (vvRB1B) strains. Using the RNA sequencing approach, we analyzed the generated transcriptomes from lymphocytes isolated from chicken bursa and spleen. Our findings validated the expression of previously characterized genes; however, they also revealed the expression of novel genes during the MDV-mediated immune response. The results showed that after challenge with CVI988 or vvRB1B strains, 634 and 313 differentially expressed genes (DEGs) were identified in splenic lymphocytes, respectively. However, 58 and 47 DEGs were observed in bursal lymphocytes infected with CVI988 and vvRB1B strains, respectively. Following MDV CVI988 or vvRB1B challenge, the bursal lymphocytes displayed changes in IL-6 and IL-4 gene expression. Surprisingly, splenic lymphocytes exhibited an overwhelming alteration in the expression of cytokines and cytokine receptors involved in immune response signaling. On the other hand, there was no distinct trend between infection with CVI988 and vvRB1B and the expression of cytokines and chemokines, such as IL-10, IFN-γ, STAT1, IRF1, CCL19, and CCL26. However, the expression profiles of IL-1β, IL-6, IL8L1, CCL4 (GGCL1), and CCL5 were significantly upregulated in splenic lymphocytes from chickens infected with CVI988 compared with those of chickens infected with vvRB1B. Because these cytokines and chemokines are considered to be associated with B cell activation and antigenic signal transduction to T cells, they may indicate differences of immune responses initiated by vaccinal and virulent strains during the early phase of infection. Collectively, our study provides valuable data on the transcriptional landscape using high-throughput sequencing to understand the different mechanism between vaccine-mediated protection and pathogenesis of virulent MDV in vivo.

## 1. Introduction

Marek’s disease (MD) is a lymphoproliferative disease that is devastating for the poultry industry as it causes huge monetary losses (1–2 billion US dollars) annually [1]. The etiological agent, Marek’s disease virus (MDV), is an alpha herpes virus that belongs to the family *Herpesviridae*, has tropism for immune cells (B and T cells) in vivo, and is widely known to cause oncogenic immunosuppression and lymphoma in chickens [1,2]. The MDV infection is initiated by inhalation of airborne cell-free virions present in contaminated dust and dander, which are continuously shed from the epithelium of feather follicles of the infected host. Upon arrival in the respiratory epithelia, MDV is transported via macrophages to the lymphoid organs, including the bursa of Fabricius (from now on referred to as the bursa in this manuscript for the sake of clarity) and the spleen, where its replication takes place [3,4,5]. The MDV life cycle in birds comprises four phases. The early cytolytic phase (2–7 days post-infection (dpi)) is designated as the semi-productive MDV mediated lytic infection in B cells [6,7]. The second phase is termed the latent phase, and develops 7–10 dpi in T cells, which become infected and activated. At 18 dpi, the third phase begins, which involves immunosuppressive and lytic MDV infection, where MDV reactivation in CD4+/CD8− T cells take place. The fourth phase is a tumorigenic proliferative phase around 28 dpi and is broadly associated with tumor formation [8,9]. Consequently, B and T lymphocytes are the main target cellular population in the early and latent stages of viral infection [10]. Meanwhile, B cells not only provide a suitable site for the expression of viral antigens, but also may be responsible for presenting antigenic signals to T cells, so as to break immune tolerance and establish a lasting antigen-specific T cell response. The enhanced susceptibility to other pathogens including coccidiosis and *E. coli* has been previously reported during the early cytolytic infection [11,12]. Studies have shown that the response to vaccination against other pathogens (*Mycoplasma synoviae*, infectious bronchitis, tetanus, etc.) is also compromised in this phase. On the basis of patterns of virulence, studies have also reported transient atrophy of lymphoid organs during MDV infection [13,14]. The bursa and thymus are considered to be the primary sites for the development of B and T lymphocytes [15]. Therefore, the activation of B cells is crucial in order to achieve a long-term antigen-specific T cell immune response [16]. In the early stage of infection (2–7 dpi), MDV replication is maximum in splenic lymphocytes [17]. Another study also documented that splenic B cells are crucial in MDV replication, and a high MDV titer is due to the replicating virions within B cells [2].

In the past few decades, viral evolution has resulted in the emergence of strains with increasing virulence, from mild to virulent (v), very virulent (vv), and very virulent+ (vv+), which reflect the levels of pathogenicity [18]. Despite massive outbreaks, vaccination has been the primary strategy in the prevention and control of MD, since the first MD vaccine was prepared in the late 1960s [19,20]. The MD vaccines have undergone several improvements and modifications over time with increasing virulence of MDV [21,22]. The most commonly used vaccinal strains include HPRS-16, SB-1, FC-126, and CVI988/Rispens, as well as a few recombinant viruses, and meq gene-deleted strains such as rMd5∆meq. All of these vaccines demonstrated a significant role in reducing MDV replication. In particular, CVI988/Rispens (a naturally-attenuated vMDV strain that establishes infection and lifelong immunity, effectively protecting against tumors and mortality) is known to be the most effective vaccine virus strain [20,23,24,25]. However, the MDV-mediated immune response, and in particular, how these vaccines activate the host protective immune response during the early cytolytic phase of infection, is still poorly studied.

In this study, using the deep sequencing approach, we investigated and elucidated the cellular and molecular mechanisms associated with vaccinal (CVI988) and natural (vvRB1B) strains of MDV in the chicken body (in vivo). Our results demonstrate that significant differences are present in the transcriptional profile of the CVI988- and vvRB1B-mediated immune responses in splenic lymphocytes at the early cytolytic stage.

## 2. Materials and Methods

### 2.1. Ethics Statement

All animal experiments were conducted in complete accordance with the Chinese Regulations of Laboratory Animals Guidelines (Ministry of Science and Technology of People’s Republic of China) and Laboratory Animal-Requirements of Environment and Housing Facilities (GB14925-2010, National Laboratory Animal Standardization Technical Committee). The license number associated with this research protocol was 2017-0039, ID: SYXK(Jing), which was approved by the animal welfare committee of Beijing Academy of Agricultural and Forestry Sciences (15 December 2017).

### 2.2. Experimental Animals and Design

All animal experiments were conducted in separate compartments of the temperature-controlled experimental animal house. Chicks were acquired from specific-pathogen-free (SPF) flocks of the White Leghorn lines, maintained at Beijing Boehringer Ingelheim Vitao Biotechnology Co., Ltd., China. One hundred and five SPF chicks (mixed-sex and one-day-old) were divided randomly into three groups (35 chickens in each group) and each group was reared in separate cages. The birds were provided with ad libitum feed acquired from CP Group Ltd., P.R. China and unrestricted free access to drinking water was given by nipple drinkers. Each bird in the treatment group was challenged via subcutaneously injecting 5000 plaque forming units (pfu) of vvRB1B or CVI988/Rispens prepared from chicken embryo fibroblast cell (CEF) suspension (0.2 mL) in the neck. The mock-treated chicks in the control group were inoculated in the neck subcutaneously with 10^5^ CEF (0.2 mL). Chicks were euthanized humanely and samples from the bursa and spleen were collected from five birds of each group (control, CVI988, RB1B) at regular intervals (1, 2, 3, 4, 5, 6, and 7 dpi).

### 2.3. Separation of Lymphocytes and Extractions of Total DNA

Splenocytes and bursal cells were harvested [26] in 5 mL phosphate buffer saline (PBS). Briefly, these were underlaid with 5 mL of Histopaque 1077 (Sigma Aldrich, St. Louis, MO, USA) in a 15 mL tube, and centrifuged at 672 g for 20 min at room temperature. They were harvested from the interface of the Histopaque (Sigma Aldrich, St. Louis, MO, USA) and PBS, and then washed with cold PBS three times at 4 °C. After cell counting, the total DNA that was used for quantification of the virus by real-time PCR was extracted from the lymphocytes (10^6^ cells) following the set guidelines described by the Qiagen Blood & Cell Culture DNA Maxi Kit (Qiagen, Venlo, Netherland). The DNA concentration of each sample was quantified and adjusted to 50 ng/μL using the NanoDrop spectrophotometer (ThermoFisher, Waltham, MA, USA).

### 2.4. Absolute Quantitation of MDV in Chicken Lymphocytes Using Real-Time PCR

MDV specific bacterial artificial chromosome (BAC) DNA was used as the standard for the establishment of the standard curve equation for the absolute quantitation of the Meq genes. In order to quantify the copy number for the Meq gene in the plasmids, we performed serial dilutions ranging from 10^8^ to 10^1^ copies/2 μL of CVI988-BAC [27] as standard controls. The previously reported primers of the Meq gene were employed and the primer sequences are presented here as follows: forward primer: 5′-CCCAACAGCCCCTCCAAACAC-3′ and reverse primer: 5′-CTTCATGGAGTTTGTCTACA-3′ [28], and the Meq gene primers were designed to amplify a conserved region that is common to both CVI988 and vvRB1B. SYBR Green qRT-PCR was performed in a Bio-Rad Real-Time PCR Detection System (Bio-Rad, CFX connect, USA), under the following conditions: pre-incubation and denaturation at 95 °C for 2 min, followed by 40 cycles of 95 °C for 15 s and 58.6 °C for 1 min.

### 2.5. High-Throughput RNA-Sequencing

Briefly, total RNA that was used for high-throughput sequencing was extracted from lymphocytes using TRIzol Reagent (Life Technologies; Carlsbad, CA, USA) following the manufacturer’s instructions. Moreover, the purity and fragment length of RNA was assessed using the NanoDrop spectrophotometer (Peqlab, USA) and Agilent 2100 Bioanalyzer (Agilent Technologies; Böblingen, Germany), respectively. Additionally, RNA was enriched by using Oligo (dT) magnetic beads and cleaved into short fragments. Subsequently, the first single strand of cDNA was synthesized by using random hexamer primers, and the double-stranded cDNA was synthesized by using buffers, dNTPs, and DNA polymerase I. Then, AMPure XP beads (Beckman Coulter, Krefeld, Germany) were used to purify and select the size of double-stranded cDNA. Finally, PCR amplification was performed to construct the cDNA library. The insertion fragment size and the effective concentration were detected and accurately quantified by Agilent 2100 Q-PCR to achieve the highest quality library standards. RNA-seq was obtained by using the Illumina high-throughput sequencing platform (HiSeqTM4000). The acquired reads were aligned to the reference genome database (ftp://ftp.ensembl.org/pub/release-92/fasta/gallus_gallus/dna/). Tophat2 software was utilized to compare the obtained sequence with the genome reference sequence. Fragments per kilobase of exon model per million mapped reads (FPKM) of each gene were analyzed on the basis of length of the gene and read count mapped to this gene. Differential expression analysis was accomplished by employing DESeq [29,30].

### 2.6. Bioinformatics Analysis of Differentially Expressed Gene Sets

GO (gene ontology, http://www.geneontology.org/) is an international standardized gene functional classification system. It aims to establish a language vocabulary standard that can be used to define and describe the functions of genes and proteins, which can be applied to all species. GO is divided into three ontologies: molecular function, biological process, and cell component. The basic unit of GO is a term, and each term corresponds to a function or attribute. In this study, Goseq is the software used for GO enrichment analysis. To highlight the most relevant GO terms associated with the DEGs, we performed GO term enrichment analysis by Goseqsoftware (v1.22). KEGG (Kyoto Encyclopedia of Genes and Genomes, http://www.kegg.jp) is a database for systematic analysis of gene functions and genomic information. It helps researchers study genes and expression information as a whole network. It not only provides all possible metabolic pathways but also comprehensively annotates the enzymes that catalyze each step of the reaction, including the amino acid sequence, PDB library link, and so on. It is a powerful tool for metabolism analysis and metabolism network research in the organism. In vivo, different genes coordinate with each other to perform their biological functions. The most important biochemical metabolic pathway and signal transduction pathway involved in differential expression genes can be determined through pathway significant enrichment. The significance enrichment analysis of the pathway was based on the unit of the pathway in the KEGG database. The hypergeometric test was used to find the significant enrichment pathway in the differential expression genes compared with the whole transcriptome background. The pathway of DGEs found in this study was analyzed by KOBAS software (v2.0).

### 2.7. Relative Quantification of Differentially Expressed Genes in Chicken Lymphocytes by Quantitative Reverse Transcriptase PCR (RT-qPCR)

Total RNA that was used for detecting DGEs was extracted from the lymphocytes of spleen and bursa using the RNeasy Mini Kit (Qiagen, Venlo, Netherland). The RNA was quantified and reverse transcribed into cDNA utilizing the FastKing RT Kit (with gDNase) according to the manufacturer’s instructions (Tiangen Biotech (Beijing) Co., Ltd.). Results were documented as target gene expression levels normalized to the glyceraldehyde-3-phosphate dehydrogenase (GAPDH) gene. A detailed list containing primers for target genes and GAPDH is shown in Table 1. The quantitative PCR (qPCR) was performed with the following conditions: initial denaturation was performed at 95 °C for 1 min, followed by 40 cycles of denaturation at 95 °C for 15 s and annealing/extension at 60 °C for 1 min, with endpoint melting-curve analysis.

### 2.8. Statistical Analysis

All the tests were performed in triplicate. Statistical differences between the treated and control groups were determined and analyzed by analysis of variance (ANOVA) using SPSS software, version 18.0 (SPSS, Chicago, IL, USA). Graphs were generated using GraphPad Prism 5.0. The statistic estimates are shown as means ± SD of three independent experiments per chicken and per indicated organ. “NS” indicates no significant difference (*p* > 0.05). “*”, “**” and “***” indicate statistically significant differences with values of *p* < 0.05, *p* < 0.01, and *p* < 0.001, respectively. The *p* value is corrected by the method of Benjamini and Hochberg in order to control the false discovery rate (FDR) [31].

## 3. Results

### 3.1. The Differences in MDV Viral Replication between a Vaccine Strain and Virulent Strain in Lymphoid Organs of the Chicken

In order to determine the MDV replication kinetics during the early stage of infection, lymphocytes were isolated from the spleen and bursa of chicks from 1–7 dpi with CVI988 or vvRB1B strains. The RT-qPCR based on the Meq gene was employed for calculation of MDV genome copy numbers in the DNA extracted from lymphocytes from infected chickens. Our data showed that on 5 dpi, the viral replication reached its peak in splenic and bursal lymphocytes (Appendix A), which was in agreement with the previously reported literature [17]. Additionally, our results signified that the genome copy numbers of both MDV strains (CVI988 and vvRB1B) in splenic lymphocytes were higher than in bursal lymphocytes (Figure 1A,B). However, the genomic copies of vvRB1B were found to be higher than those of CVI988 in both splenic and bursal lymphocytes (Figure 1C,D).

### 3.2. Differentially Expressed Genes (DEGs) in Splenic Lymphocytes Post-MDV Infection

To primarily evaluate the host immune response to MDV infection during the early cytolytic stage, we analyzed the differences in gene expression between lymphocytes from chickens pre-infected with CVI988 and lymphocytes from chickens infected with vvRB1B (treatment group) at 5 dpi as well as uninfected lymphocytes (control group) by using the RNA-seq approach. Considering uninfected lymphocytes as a control, fewer differentially expressed genes (DEGs) were observed in bursal lymphocytes compared to those in splenic lymphocytes from chickens infected with either MDV CVI988 or vvRB1B (Appendix A). Moreover, there were more DEGs in splenic lymphocytes from chickens infected with MDV CVI988, compared to those from chickens infected with MDV vvRB1B (Figure 2), according to the statistical criteria of *p* < 0.05 and |log2fold change > 1.5. Among these DEGs, 408 up-regulated and 226 down-regulated DEGs were identified between the cells from chickens infected with CVI988 and cells from uninfected chickens (Figure 2A), but there were 189 up-regulated and 124 down-regulated DEGs identified in the cells from chickens infected with vvRB1B and cells from uninfected chickens (Figure 2B). Furthermore, there were 158 up-regulated genes conserved in the cells from chickens infected with CVI988 and cells from chickens infected with vvRB1B, and 250 up-regulated genes were specific for CVI988, whereas only 31 up-regulated genes were specific for vvRB1B (Figure 2C). Correspondingly, there were 61 down-regulated genes shared by CVI988 and vvRB1B, and 165 down-regulated genes were specific for CVI988 and only 63 down-regulated genes were specific for vvRB1B (Figure 2C). Collectively, the DEGs of lymphocytes infected with CVI988 were much more numerous than those of lymphocytes infected with vvRB1B in the spleen.

Among these up-regulated genes, the encoded products are involved in the host immune response, including interferon type I (IFNα) and interferon type II (IFN-γ), some interleukins (IL-6, IL-10), the chemokine CCL19, genes involved in the interferon response and controlling the interferon response (IRF1, BATF3, STAT1, IFIH1, SOCS1, and SOCS3), immunoresponsive gene 1 (IRG1, also termed ACOD1), the proinflammatory protease granzyme A (GZMA), and the avidin gene (AVD), which had been identified in previous studies [32,33,34,35]. In addition to the aforementioned genes, several other genes associated with the innate immune response were also upregulated, including colony-stimulating factor 3 (CSF3), the chemokines IL-8, IL8L1, CCL4, and CCL5, inducible nitric oxide synthase (NOS2), interleukins and interleukin receptors (IL-19, IL-22, IL1R2, and IL-18RAP), and genes involved in governing the interferon response (IFI6, IFI27L1, and IFIT5). Importantly, the expressions of IL-6, IL-1β, IL8L1, CCL4, and CCL5 were higher in lymphocytes from chickens infected with CVI988 compared with lymphocytes from chickens infected with vvRB1B (Figure 4). Interestingly, the down-regulated genes in this study were not related to immune responses.

### 3.3. Gene Ontology (GO) Terms and Kyoto Encyclopedia of Genes and Genomes (KEGG) Pathways of DEGs Associated with the Immune Response in Splenic Lymphocytes after MDV Infection

KEGG distinctly segregates the DEGs on the basis of functionality, whereas GO hierarchically distributes the cohort of genes into three distinct classes of cellular, molecular, and biological terms, respectively. To determine the functionality associated with DEGs, analysis of GO terms and KEGG pathways was conducted. The results showed that there were more up-regulated DEGs associated with the immune response to CVI988 than to vvRB1B in light of uninfected lymphocytes, which significantly enriched biological molecules related to the immune system process, immune response, gas transport, and oxygen transport (Figure 3A,B). The notable molecular functions of the DEGs in the CVI988 group consisted of cytokine activity, cytokine receptor binding, receptor binding, G-protein coupled receptor binding, chemokine activity, and chemokine receptor binding (Figure 3A). However, the remarkable molecular functions for the DEGs in the vvRB1B group included cytokine activity, receptor binding, protein binding, and trypsin activity (Figure 3B), and the genes for cytokine activity of molecular function in the vvRB1B group were not significantly different from those in the CVI988 group (Figure 3A,B). The results also indicated that CVI988 stimulated more interaction of cytokines with cytokine receptors and chemokines with chemokine receptors compared to vvRB1B, but no down-regulated DEGs were significantly enriched in processes of the immune response to CVI988 and vvRB1B infection (Figure 3C,D).

In addition, the analysis of KEGG pathways revealed that the host response pathways were also considerably (corrected *p* value < 0.25) affected during the host response to CVI988 and vvRB1B infection, respectively, including cytokine–receptor interactions, the NOD-like receptor, RIG-I-like receptor signaling, and the Toll-like receptor signaling pathway, which are associated with innate as well as adaptive inflammatory host defenses, cell growth, differentiation, and cell death (Figure 3E,F).

### 3.4. Cytokines and Chemokines Differentially Expressed in Splenic Lymphocytes after Infection with MDV

To learn which cytokines and chemokines mediated the immune response to MDV, RT-qPCR was employed to quantitatively measure mRNA transcription of selected immune factors encoding genes, including 13 up-regulated genes IL-1β, IL-6, IL-10, IFN-γ, STAT1, IRF1, IL8L1, CCL4, CCL5, CCL19, CCL26, and one down-regulated gene (IRF5). RT-qPCR results validated the gene expression of all tested immune factors from RNA-seq data. Meantime, RT-qPCR also verified that CVI988 stimulated higher expression levels of IL-6, IL-1β, IL8L1, CCL4, and CCL5, compared with vvRB1B, although IL-10, IFN-γ, STAT1, IRF1, CCL19, and CCL26 were up-regulated in spleen lymphocytes from chickens infected with either CVI988 or vvRB1B (Figure 4). In addition, the levels of IL-4 and IL-6 increased significantly in bursal lymphocytes infected with vaccine and virulent MDV (Appendix A). The protein–protein interaction (PPI) analysis, performed by using the STRING (https://string-db.org/) database, showed that CCL4 is involved in the biological interaction network of the host response to MDV in the spleen at 5 dpi, as it was highly interconnected with other chemokines, such as CCL5, CCL19, CCL26, and IL8L1 (Figure 5A). This finding was consistent with previous work (Figure 5B).

## 4. Discussion

MD is an excellent disease model for studying viral oncogenesis, and the MD vaccine was documented as the first vaccine capable of protecting the host against tumors induced by an oncogenic virus. MDV mediated pathogenesis is complex and the ability of MDV to induce atrophy in the thymus and bursa during the early phase of infection has been previously reported [36]. The virulence of an MDV strain determines the degree of atrophy in the lymphoid organs [18]. Therefore, it is imperative to explore and compare the patterns of immune responses after infection with vaccinal and virulent MDV strains in vivo.

Like other oncogenic herpes viruses, the course of MDV infection comprises four phases: early productive replication, establishment of latency, reactivation to produce viral progeny, and the transformation phase [37]. The early cytolytic phase of MDV occurs mainly in B lymphocytes of the spleen and bursa within 2–7 dpi [7]. What’s more, MDV-antigen positive cells are mainly B lymphocytes (around 90%), while CD4+ and CD8+ lymphocytes account for only 3% and 8%, respectively, in the early cytolytic stage [2,38,39]. MDV infects and causes alterations in proliferation and activation of B cells. Studies have shown that MDV affects the disease outcome by merely altering or controlling the host cellular machinery in order to facilitate its replication [40]. Meanwhile, it is the innate immune response that is chiefly involved in the battle against invading MDV. Therefore, MDV replication in the host cells, together with the MDV progression, mainly depends on a certain set of interactions, including the virus–host interactions. To date, studies have mainly elaborated on the remarkable viral virulence factors contributing to the course of disease progression. However, using advanced tools of high-throughput sequencing, the dynamic transcriptome of MDV-mediated immune responses is not fully identified. Thus, the present study was conducted to investigate the transcriptional variations and the immunological differences in the early phase of MDV-mediated infection in the lymphocytes of the spleen and bursa from chickens infected with either a vaccinal strain or a virulent strain. The present study deeply examined the expression of specific chemokines, cytokines, interferon-stimulated genes (ISGs), and signaling pathways related to the immune response. It will bring new horizons to understanding the MDV-mediated immune response in the lymphoid organs of the chicken.

Firstly, we evaluated the replication kinetics of MDV in lymphocytes of lymphoid organs (spleen and bursa) during a specified time (1–7dpi) in chickens infected with MDV CVI988 or vvRB1B strains and found that the MDV replication reached its peak in splenic and bursal lymphocytes on the fifth day. Interestingly, splenic lymphocytes possessed a higher viral load than bursal lymphocytes. Fascinatingly, the replication kinetics revealed that the vvRB1B replicated faster than CVI988 (Figure 1A,B), which is consistent with the previously reported experimental results [17]. In addition, this phenomenon may be associated with the B lymphocytes. Previous research has shown that the high titer of circulating virus is mainly because of the replication of the virus within B cells in spleen [2]. Due to the fact that CVI988 and vvRB1B are the representatives of vaccines and very virulent strains, respectively, the difference in viral replication of CVI988 and vvRB1B implied that the early targets (B and T lymphocytes in spleen/bursa) undergo distinct evasion.

Utilizing genome-wide high-throughput sequencing (RNA-seq), our main focus was to determine the transcriptional changes associated with the MDV-mediated immune response. The results revealed that at 5 dpi, fewer genes were differentially expressed in bursal lymphocytes than those in splenic lymphocytes from chickens infected with either MDV CVI988 or vvRB1B (see Appendix A). This suggested that splenic lymphocytes exerted a potent immune response against the infection of MDV, which might be due to the fact that the degree of host immune response is negatively correlated with the viral load of MDV. In the chicken, due to the lack of lymph nodes, the spleen is the primary site where the immune response towards systemically administered antigens is located [41]. Moreover, the spleen contains mainly B and T lymphocytes that are the major targets of MDV. It can be postulated that the spleen is the main battlefield where immunocytes combat the invading MDV in the early stages. Moreover, compared to vvRB1B infection, the upregulation of distinct sets of genes was seen in response to CVI988 treatment in splenic lymphocytes (Figure 2). This suggests that, in response to the vaccinal strain of MDV, a more potent antiviral environment is elicited in the cells that further validate the functionality and efficiency of the MDV vaccine.

Alteration of biological pathways is coincident with expression patterns of the signal molecules during the host response to infection, so the DEGs were further analyzed based on the gene ontology (GO) and Kyoto Encyclopedia of Genes and Genomes (KEGG) pathways [42]. The KEGG analysis showed that, compared with vvRB1B, the NOD-like and RIG-I-like receptor signaling were among the differential pathways related to CVI988, and it also showed that CVI988 could initiate the innate immune response, and then induce an adaptive immune response (Figure 3E,F).

Furthermore, compared to vvRB1B infection, the significant upregulation of interleukins and pro-inflammatory cytokines, including IL-1β and IL-6 under the influence of CVI988 (Figure 4) further postulated that the proliferation of Th1 cells under the influence of IL-6 and IL-1β might be a characteristic of the MDV vaccine in spleen. These mediators are broadly associated with apoptosis, cellular proliferation and differentiation, inflammation, cytokine–cytokine receptor interactions, and the antiviral response [43]. However, the upregulation of interleukins, especially IL-4 and IL-6 were markedly enhanced in the bursal lymphocytes in response to both virulent and vaccinal MDV (Appendix A). Th2 cells are the main source of IL-4 secretion and play a crucial role in B cell activation. On the other hand, IL-6 is a soluble mediator with a pleiotropic effect on inflammation, immune responses, and hematopoiesis. Based on its ability to functionally differentiate the B cells into the active antibody (Ab)-producing cells, IL-6 is also termed the B-cell stimulatory factor 2 (BSF-2) [44,45]. It is evident from experimentation that IL-6 stimulated the secretory activity of B cells during Epstein–Barr virus (EBV) infection [32,46]. Similarly, the same phenomenon was observed in our study where the upregulation of IL-6 distinctly demonstrated its crucial immunomodulatory role in a MDV-mediated immune response.

IL-10 has multiple effects on immune cells. It has been shown to stimulate the proliferation and cytotoxic activity of natural killer cells, as well as the proliferation and cytolytic activity of CD8+ T cells in vitro in the presence of IL-2, and to enhance B cell proliferation and differentiation [47,48]. IFN-γ signaling can activate JAK/STAT1, MAPK, and NF-kappa B signaling pathways to regulate the expression of a number of genes, which play a key role in host defense by promoting macrophage activation, upregulating the expression of antigen processing and presentation molecules, driving the development and activation of Th1 cells, regulating B cell functions, and inducing the production of chemokines that promote effector cell trafficking to sites of inflammation [49,50]. The up-regulation of IL-10, IFN-γ, and STAT1 in this article suggests that MDV infection is associated with the proliferation and differentiation of lymphocytes in spleen.

Chemokines are secretory chemotactic cytokines that play crucial roles in biological processes, including angiogenesis, inflammation, cellular recruitment, and lymphatic organ development, thus controlling the immune response [51,52]. CCL4 is also known as macrophage inflammatory protein-1β (MIP-1β) because CC chemokines possess specificity for CCR5 receptors. It is evident from the literature that CCL4 is a major HIV-suppressive factor produced by CD8+ T cells [53] and is involved in the recruitment and transportation of the lymphoid cells. However, the possible role of CCL4 in avian immune signaling has not been fully determined. The significant upregulation of CCL4 in response to CVI988 infection in lymphocytes further postulates that CCL4 might be a fundamental gene in MDV-mediated early cytolytic infection. Further experimentation and functional verification are mandatory to broadly characterize the role of CCL4 in the cascade of the chicken immune response. Like its mammalian counterpart, CCL4 in chickens may have similar roles in the proliferation, differentiation, recruitment, and migration of immune cells [54]. Similarly, CCL5, being an active member of the CC subfamily, performs the same functions as CCL4, including being a chemoattractant for blood monocytes, memory T helper cells, and eosinophils [53]. CCL19, another cytokine, plays a significant role in immune trafficking, cellular migration, and differentiation in secondary lymphoid organs [55]. CCL26 displays chemotactic activity for normal peripheral blood eosinophils and basophils. This chemokine may contribute to the eosinophil accumulation in atopic diseases [56,57]. Collectively, the upregulation of the aforementioned chemotactic cytokines implies that they play a crucial role in the migration of lymphoid cells to the target tissue and are solely involved in innate immune signaling in the chicken. IL8L1 is another chemokine encoded only by chickens, and its specific functions need to be further explored. However, IL8L1 probably associates with vIL8, which recruits T cells to B cells infected with MDV and leads to transduction of viral signal from B to T cells [58]. The upregulation of the above-mentioned chemokines in the present study under the influence of MDV infection, particularly CVI988 infection, further validates the role of these potential cytokines in MDV-mediated immune responses. Furthermore, these findings strongly documented that a stronger antiviral environment is established in response to CVI988 compared to vvRB1B, in immune organs of chicken during the early cytolytic phase of infection. However, this warrants future inclusive studies to broadly characterize and delineate the MDV-mediated immune response during early cytolytic infection.

In conclusion, using the advanced tools of sequencing (RNA-seq), we presented a comparative scenario in the MDV-mediated immune response in the lymphoid organs of chickens during the early phase of infection. We identified the highly significant genes associated with the host’s immune competence and the results showed that lymphocytes from CVI988-infected chickens could elicit a stronger early immune response, compared to lymphocytes from vvRB1B-infected chickens. We proposed the hypothesis that early vaccination with the CVI988 strain may halt the establishment of latent infection with very virulent MDV. Herpes virus is peculiar in the establishment and maintenance of latency post cytolytic infection. The present study was designed and performed to broadly examine the MDV-mediated immune response, and characterization of differentially expressed genes in response to CVI988 and vvRB1B MDV strains by using high-throughput RNA sequencing. The findings of the present study will provide a deeper insight into the MDV-mediated immune response in the chicken. The MD pathobiology and the possible role of DEGs in the cascade of MDV-mediated immune signaling need further extensive investigations involving the functional characterization of DEGs both in vivo and in vitro.

## Figures and Tables

**Figure 1 viruses-12-00354-f001:**
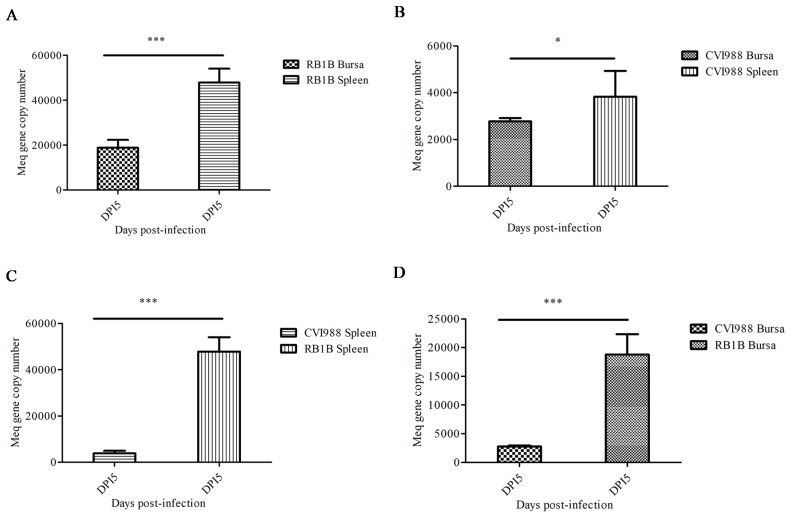
Replication of vvRB1B and CVI988 strains of Marek’s disease virus (MDV) in splenic and bursal lymphocytes. (**A**) The viral load in vvRB1B infected splenic and bursal lymphocytes. (**B**) The viral load in CVI988 infected splenic and bursal lymphocytes. (**C**) The comparison of replication of vvRB1B and CVI988 in bursal lymphocytes. (**D**) The comparison of replication of vvRB1B and CVI988 in splenic lymphocytes. The statistic estimates are shown as means ± SD of three independent experiments of per chicken and per indicated organ (* *p* < 0.05; *** *p* < 0.001; ns, no significance).

**Figure 2 viruses-12-00354-f002:**
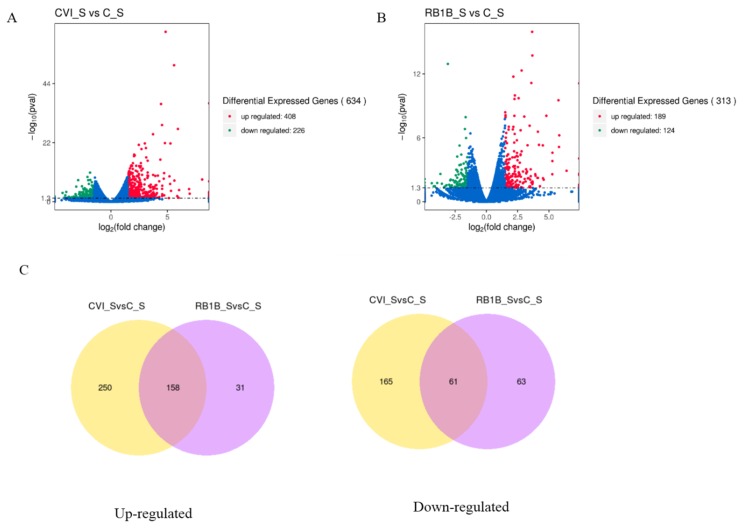
The overall distribution of differentially expressed genes (DEGs) presented by the volcano plot and Venn diagrams. Three individuals in each group were used for identifying DEGs. (**A**) DEGs in splenic lymphocytes from chickens infected with CVI988 compared to the control group. (**B**) DGEs in splenic lymphocytes from chickens infected with vvRB1B compared to the control group. Red, green, and blue dots represent the upregulation, downregulation, and total DEGs, respectively. (**C**) Venn diagrams representing the genes in common among treatment groups. (The statistical criteria are *p* < 0.05 and log2fold change > 1.5).

**Figure 3 viruses-12-00354-f003:**
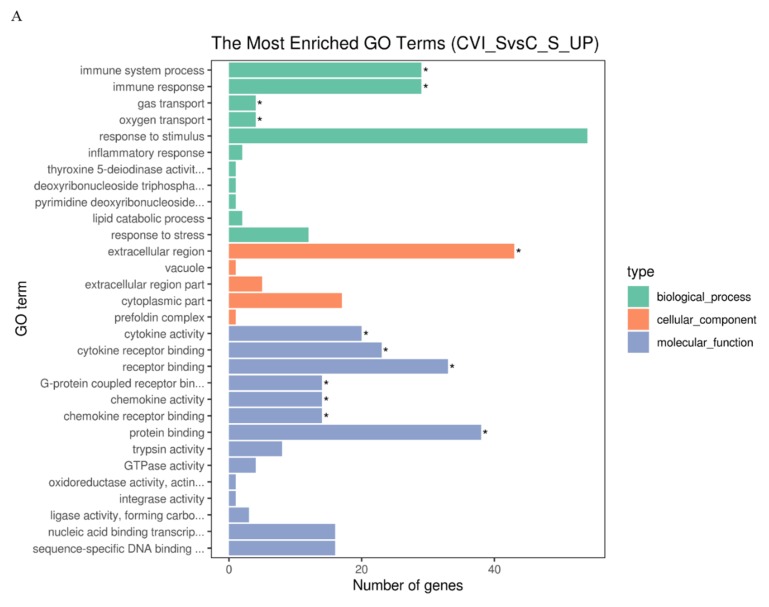
Gene Ontology (GO) terms and KEGG pathway enrichment of the DEGs. The most enriched GO terms for up-regulated DEGs in CVI988 versus control (**A**) and vvRB1B versus control (**B**), down-regulated DEGs in CVI988 versus control (**C**), and vvRB1B versus control (**D**). The most enriched KEGG pathway for up-regulated DEGs in CVI988 versus control (**E**) and vvRB1B versus control (**F**). (* indicates significant enrichment of GO term; q value (range 0–1) is the *p* value that was corrected by multiple hypothesis tests. The closer the q value is to zero, the more significant the enrichment of the KEGG pathway).

**Figure 4 viruses-12-00354-f004:**
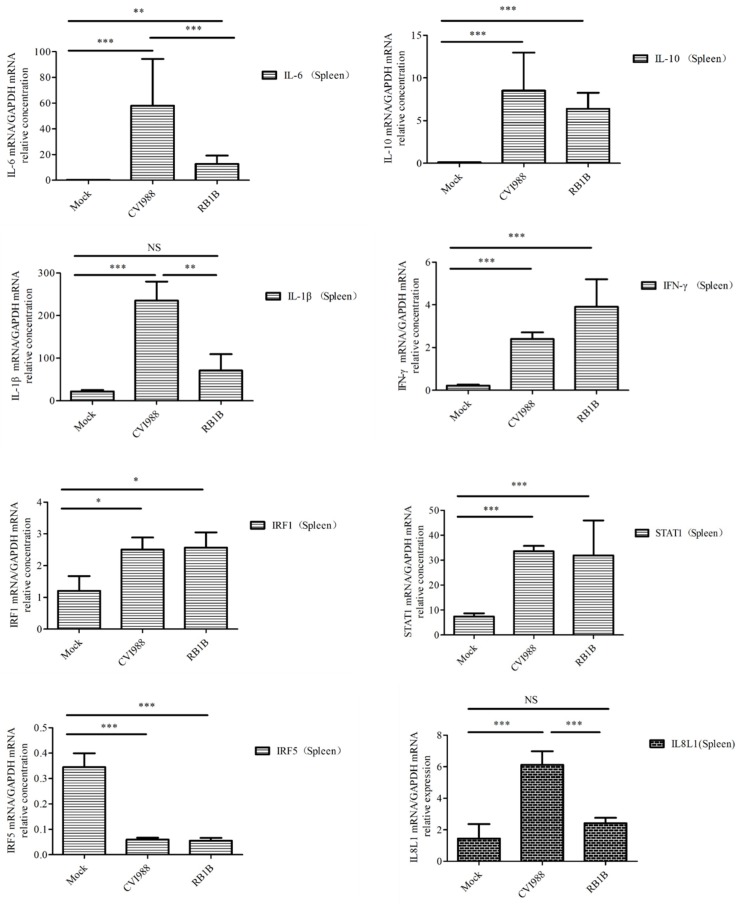
Analysis of DEG expression of splenic lymphocytes in CVI988, vvRB1B, and mock groups. The relative quantity of IL-6, IL-1β, IL-10, IFN-γ, STAT1, IRF1, IL8L1, GGCL1(CCL4), CCL5, CCL19, and CCL26 (CCLI10) gene expression of splenic lymphocytes in CVI988, vvRB1B, and control groups. “NS” indicates no significant difference (*p* > 0.05). “*”, “**” and “***” indicate statistically significant differences with values of *p* < 0.05, *p* < 0.01, and *p* < 0.001, respectively.

**Figure 5 viruses-12-00354-f005:**
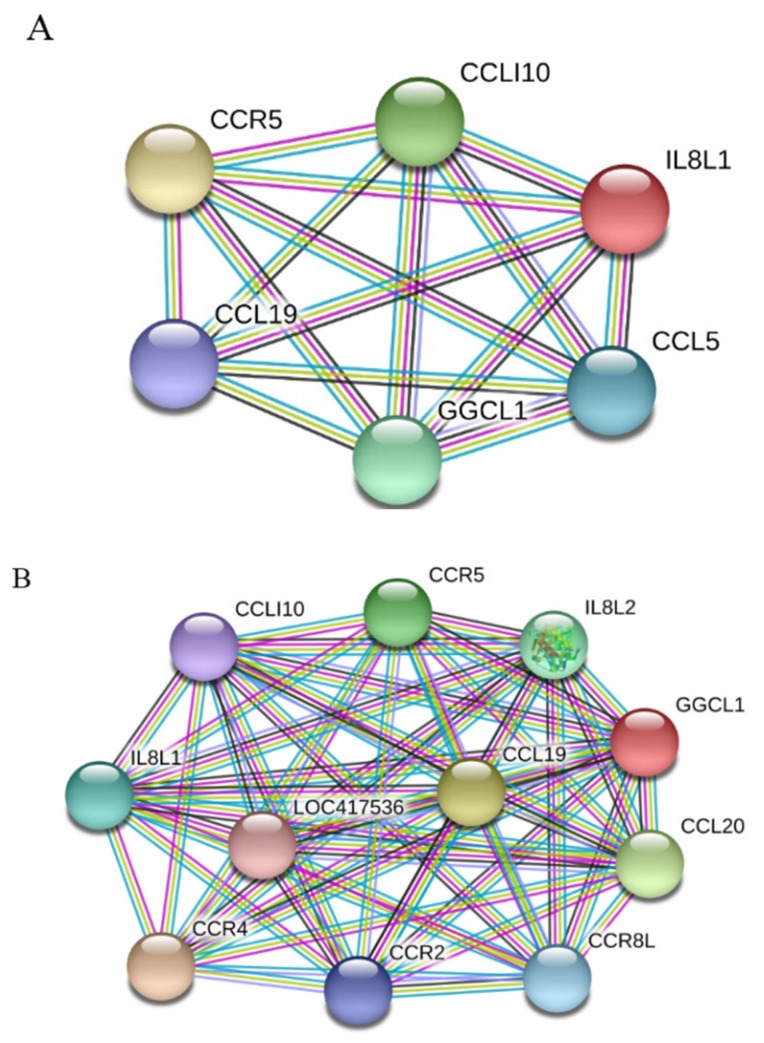
Protein–protein interaction (PPI) network analysis obtained using STRING. (**A**) Network representing genes that interact with CCL4 in response to MDV infection. (**B**) Genes currently known to interact with CCL4. Differently colored lines represent different types of evidence used in predicting interactions. Known interactions, light blue line: from curated databases; purple line: experimentally determined. Predicted interactions, green line: gene neighborhood; red line: gene fusion; blue line: gene co-occurrence. Others, yellow line: text mining; black line: co-expression; light purple line: protein homology.

**Table 1 viruses-12-00354-t001:** List of primers used in this study.

Primers ^*^	Sequence (5′-3′)	Amplicon Size (bp)
GAPDH-F GAPDH-R	AGAACATCATCCCAGCGT AGCCTTCACTACCCTCTTG	182 bp
IL-6-F IL-6-R	AAATCCCTCCTCGCCAATCT CCCTCACGGTCTTCTCCATAAA	106 bp
IL-4-F IL-4-R	AGCACTGCCACAAGAACCT GCTAGTTGGTGGAAGAAGGTAC	140 bp
IL-10-F IL-10-R	GCTGCGCTTCTACACAGATG TCCCGTTCTCATCCATCTTC	203 bp
IL-1β-F IL-1β-R	CCCGCTTCATCTTCTACCGC GCTTGTAGGTGGCGATGTTG	159 bp
IFN-γ-F IFN-γ-R	GCCGCACATCAAACACATATCTG GCGCTGGATTCTCAAGTCGTTC	127 bp
IRF1-F IRF1-R	TGGGGTCGTCCTCAGAAGAT AGAAGCCTTTCCCCTCAACG	105 bp
IRF5-F IRF5-R	AACAAGAGCCGTGAGTTCCG CTGAAGTCATCCCCGGTCAC	122 bp
STAT1-F STAT1-R	CCAAGCGGATGGGCTTCTAT GCTGATCCGCGTTAAGTCCT	163 bp
IL8L1-F IL8L1-R	CCTCACTGCAAGAATGTGGA GGAGGAGGTAGGACGTTTTTG	153 bp
CCL4-F CCL4-R	CCTTCAGCTTTGTGGCAGAC GGTGATGAACACAACACCAGC	71 bp
CCL5-F CCL5-R	GGCTGATACAACCGTGTGCT CCTGGTGATGAACACAACTGC	121 bp
CCL19-F CCL19-R	TGTATGCTGGCAACAACGTC CACAGAGACGCTTGCCCTTT	156 bp
CCL26-F CCL26-R	TTCAGATGGCCTACCCACAAC GACTCCTCGGGGTTTACACA	162 bp

* F represents forward PCR primer; R represents reverse PCR primer.

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
