# Peer review of "Transcriptional Profiles Associated with Marek’s Disease Virus in Bursa and Spleen Lymphocytes Reveal Contrasting Immune Responses during Early Cytolytic Infection"

_viruses, 2020, doi:10.3390/v12030354_

Round 1

Reviewer 1 Report

The authors have sufficiently revised the manuscript to address my concerns.

Author Response

Point : The authors have sufficiently revised the manuscript to address my concerns.

Response : We are humble and grateful to the worthy reviewer for his suggestions on the manuscript.

Reviewer 2 Report

I have re-read the manuscript.  I am satisfied that the authors have addressed most of my comments and questions except for the two below:

  • CVI988 and RB1B have some differences in the meq gene. Were the meq gene primers designed to amplify a conserved region that is common to both CVI988 and RB1B?  Please state whether the primers detect both CVI988 and RB1B (section 2.4).
  • Figure 1: Do the values represent the mean from 5 birds? What are the bars?  Standard deviation?  Standard error?  Please mention this in the figure legend.

Figure S2 legend line 11 should refer to ‘D’ (not ‘B’).

There are some additional changes required to improve use of the English language (in the new sentences added by the authors since the last revision).  I have made these corrections in the accompanying edited PDF file.

Author Response

We are thankful to the respected reviewer for his comments on the manuscript.

Point 1: CVI988 and RB1B have some differences in the meq gene. Were the meq gene primers designed to amplify a conserved region that is common to both CVI988 and RB1B? Please state whether the primers detect both CVI988 and RB1B (section 2.4).

Response 1: We are thankful to the reviewer for his suggestions. According to the reviewer’s instructions, we have added the sentence ‘and the meq gene primers were designed to amplify a conserved region that is common to both CVI988 and vvRB1B’ in section 2.4. (Lines 148-149)

Point 2: Figure 1: Do the values represent the mean from 5 birds? What are the bars? Standard deviation? Standard error? Please mention this in the figure legend.

Response 2: We thank the reviewer’s comments. The values represent the mean from 5 birds, and the statistic estimations are shown as means ± SD of three independent experiments per chicken and per indicated organ (*p < 0.05; **p < 0.01; ***p < 0.001; ns, no significant). As suggested by the reviewer, we have added this in the figure legend. (Lines 292-294)

Point 3: Figure S2 legend Line 11 should refer to ‘D’ (not ‘B’).

Response 3: We highly acknowledge the reviewer’s suggestions. We have changed it from ‘B’ to ‘D’.

Point 4: There are some additional changes required to improve use of the English language (in the new sentences added by the authors since the last revision). I have made these corrections in the accompanying edited PDF file.

Response 4: We are highly grateful to the reviewer’s comments. In agreement with the reviewer’s suggestions. We have improved the sentences as follows:

1. We have incorporated a new sentence stating ‘Because lymphocytes of the bursa of Fabricius and spleen are prime targets of MDV replication during the early cytolytic phase of infection, the immune response in bursa and spleen should be the foundation of late immunity induced by MDV. However, the mechanism of the MDV-mediated host immune response in lymphocytes in the early stage is poorly understood’. (Lines 26-29)

2. We have changed the word ‘resulting’ to ‘has resulted’. (Line 87)

3. We have changed the sentence ‘In addition, this phenomenon may be associated with the B lymphocytes. Previous research has shown that the high titre of circulating virus is mainly because of the replication of the virus within B cells in spleen’. (Lines 385-387)

4. We have modified the sentence ‘In the chicken, due to the lack of lymph nodes, the spleen is the primary site where the immune response towards systemically administered antigens is located. And the spleen contains mainly B and T lymphocytes that are the major targets of MDV’. (Lines 398-401)

5. We have carefully revised the grammatical errors in the manuscript.

Reviewer 3 Report

The manuscript [Virus-701102], entitled “Transcriptional Profiles associated with Marek's disease Virus in Chicken B Cells Reveal Contrasting Immune responses during Early Cytolytic Infection” by Jin et al., reports finding on Marek’s disease virus (MDV)-induced transcriptomic profiles of the B cells and significant differences in expression of cytokine and chemokine genes in response to vaccine strain (CVI988) and non-vaccine strain (vvRB1B) MDV challenge.

This is an interesting study that was laid out to profile transcriptomic differences in lymphoid organs of bursa and spleen of chickens inoculated with CVI988/Rispens and RB1B MDV. The analyses of the data and the validation of the expressional data were done reasonably well. Minor changes to correct typos and format errors are pending the authors’ attention.

Some specific concerns are listed below for consideration in revision.

      Major concerns:

  1.  None.

Minor concerns:

  1. Grammatic error on line 29: Suggested to revise to “… is poorly understood.”
  2. Lines 52-53: keep the keyword format consistent.
  3. Line 89: Revise to leave a space between word, citation and punctuation.
  4. Line 106: Please correct the typo: limphoctytes (lymphocytes?).
  5. Line 205: Suggested to revise “The data …” to “The statistic estimates are shown …”
  6. Line 275: Conventionally, the verb in such sentences, for instance, “The results also indicate that …” is used in the past tense, instead of the present tense, for example “The results also indicated that …”. Suggested to revise accordingly for all such sentences. One more example: the sentence on lines 279-280 could be revised as “… the host response pathways were significantly (corrected P value < 2.05) affected during …”.
  7. Line 280: This statement claimed “Significantly (…) affected …” but the P value (corrected) was close to 0.25

Author Response

We are highly grateful to the respected reviewer for his time and comments on the manuscript.

Point : Grammatic error on Line 29: Suggested to revise to “… is poorly understood.”

Response : We are thankful to the reviewer for his precious comments. According to the reviewer’s instructions, we have corrected the sentence to ‘… is poorly understood’. (Line 29)

Lines 52-53: keep the keyword format consistent.

We have changed the words ‘Differentially Expressed Genes’ to ‘Differentially expressed genes’, ‘cytokine’ to ‘Cytokine’, ‘chemokine’ to ‘Chemokine’ in the keywords. (Lines 52-53)

Line 89: Revise to leave a space between word, citation, and punctuation.

Following your comment, we have modified accordingly.

Line 106: Please correct the typo: limphoctytes (lymphocytes?).

We have corrected the spelling mistake. (Line 106)

Line 205: Suggested to revise “The data …” to “The statistic estimates are shown …”

Following your comment, we have made the correction. (Lines 207-208)

Line 275: Conventionally, the verb in such sentences, for instance, “The results also indicate that …” is used in the past tense, instead of the present tense, for example “The results also indicated that …”. Suggested to revise accordingly for all such sentences.

Following your advice, we have corrected the sentence. (Line 278)

Line 279-280 could be revised as “… the host response pathways were significantly (corrected P value < 2.05) affected during …”.

Following your comment, we have modified the sentence. (Lines 282-283)

Line 280: This statement claimed “Significantly (…) affected …” but the P value (corrected) was close to 0.25.

Based on the corrected p-value, We have modified the sentence. (Lines 282-284)

Reviewer 4 Report

Thank you for taking the time to address my critiques and questions.  I believe that the manuscript is significantly improved and suitable for publication after editing the English language and style!   

Author Response

Point : Thank you for taking the time to address my critiques and questions.  I believe that the manuscript is significantly improved and suitable for publication after editing the English language and style!

Response : We are highly grateful to the reviewer’s suggestions. In order to improve the language, we have acquired the professional editing by MDPI's English editing services for this manuscript. We are hoping that now it will be accepted for publication in Viruses.

This manuscript is a resubmission of an earlier submission. The following is a list of the peer review reports and author responses from that submission.

Round 1

Reviewer 1 Report

This manuscript needs extensive editing for spelling, grammar and proper English phrasing. Often words are used incorrectly, for example “undermine” is misused multiple times. These errors make it difficult to follow the manuscript at times.

The biggest issue is the authors focus all of the results and discussion on B-cells. However, they did not use any type of FACS to sort the cells into type-specific fractions. This is particularly an issue the spleen data. Using Histopaque alone, will result in a mixed population of immune cells, all of which will impact the response to the virus and therefore the data. Therefore the viral load and expression differences found are not necessarily due to the B-cell response. For example on page 4 lines 176-179, the authors state that MDV copies numbers for both strains were higher in the splenic lymphocytes than bursal lymphocytes. This is most likely not due to differences in B-cell infectivity/response but most likely a reflection of the mixed population of spleen lymphocytes, which would also include infected T-cells. Also, the same is true for the differences in immune cytokines. The bursa and spleen have two different main functions. The bursa is a primary immune organ and mainly serves in B-cell proliferation and development and contains a mix of B-cells at different stages. The spleen is a secondary immune organ which serves to monitor and respond to system-wide immunological triggers and contains a myriad of immune cell types most if not all, will respond in some way to MDV. It is therefore not surprising that there was a much a larger number of differently expressed genes in the spleen than bursa and the contribution of B-cells specifically to these changes cannot be determined with the current experimental design. If the authors want the focus on the B-cells specifically then they should include an additional experiment in which they use FACS to isolate B-cells and then determine the expression of genes of interest using RT-qPCR. As is, the conclusions are not sufficiently supported by the data. The title also does not accurately reflect the data as the transcriptional profiles were not generated solely from B-cells.

Author Response

Cover letter

Ref: Viruses-701102
Title: Transcriptional Profiles Associated with Marek's Disease Virus in Bursa and Spleen Lymphocytes Reveal Contrasting Immune Responses during Early Cytolytic Infection
Journal: Viruses

Esteemed reviewer,

I am writing to you regarding the revised version of our manuscript submitted to the journal Viruses(viruses-701102). We highly appreciate you for yours time and constructive comments, which are greatly helpful for us to improve our manuscript. We have carefully revised our manuscript following the comments. One by one response to all the comments is uploaded with this revised manuscript. We would greatly appreciate your consideration for accepting our manuscript to publish in Viruses.Looking forward to your decision.

Sincerely yours

Yongqing Li

Yongqing Li

Ph.D., Professor

Research Center for Infectious Disease in Livestock and Poultry, Beijing Academy of Agricultural and Forestry Sciences

Beijing 100197, People’s Republic of China

Point 1: Title:The biggest issue is the authors focus all of the results and discussion on B-cells. However, they did not use any type of FACS to sort the cells into type-specific fractions. This is particularly an issue the spleen data. Using Histopaque alone, will result in a mixed population of immune cells, all of which will impact the response to the virus and therefore the data. Therefore the viral load and expression differences found are not necessarily due to the B-cell response. For example on page 4 lines 176-179, the authors state that MDV copies numbers for both strains were higher in the splenic lymphocytes than bursal lymphocytes. This is most likely not due to differences in B-cell infectivity/response but most likely a reflection of the mixed population of spleen lymphocytes, which would also include infected T-cells. Also, the same is true for the differences in immune cytokines. The bursa and spleen have two different main functions. The bursa is a primary immune organ and mainly serves in B-cell proliferation and development and contains a mix of B-cells at different stages. The spleen is a secondary immune organ which serves to monitor and respond to system-wide immunological triggers and contains a myriad of immune cell types most if not all, will respond in some way to MDV. It is therefore not surprising that there was a much a larger number of differently expressed genes in the spleen than bursa and the contribution of B-cells specifically to these changes cannot be determined with the current experimental design. If the authors want the focus on the B-cells specifically then they should include an additional experiment in which they use FACS to isolate B-cells and then determine the expression of genes of interest using RT-qPCR. As is, the conclusions are not sufficiently supported by the data. The title also does not accurately reflect the data as the transcriptional profiles were not generated solely from B-cells.

Response 1: We are really grateful for the reviewer’s comments. As suggested by the reviewer, the result will be more convincing if we use FACS to isolate B-cells and then determine the expression of genes of interest using RT-qPCR. However, at the present moment, there are some limitations that make this experiment impossible to perform, such as firstly, it is not easy to screen chicken B cells, and secondly, it is also problematic to screen MDV-infected chicken B cells. But beyond that, firstly, we would like to point out that the splenic lymphocytes are mainly constituted of B lymphocytes. And as we all know, in the early stage of infection, MDV mainly targets and infects B lymphocytes, especially during the early phase of infection termed as cytolytic infection in 2-7 days. Secondly, considering the fact that spleen and bursa are highly distinct in functionality, and their constituents (lymphocytes) are highly different, therefore, the response of B lymphocytes to MDV in the spleen will be different compared with that in the bursa. Thirdly, for analyzing the results of this study, we used a higher threshold(p< 0.05 and |log2fold change| > 1.5)to select the genes with extremely significant differences from a large population of DEGs (Table S1), and we speculated that there is no significant difference for the genes with extremely significant differences between mixed lymphocytes and B lymphocytes. Therefore, the results of this study can be explained by the mixed lymphocytes population. Even so, in order to avoid ambiguity, We change the title as ‘Transcriptional Profiles Associated with Marek's Disease Virus in Bursa and Spleen Lymphocytes Reveal Contrasting Immune Responses during Early Cytolytic Infection’.

Reviewer 2 Report

Viruses (MS 701102)

Transcriptional Profiles associated with Marek's disease Virus in Chicken B Cells Reveal Contrasting Immune responses during Early Cytolytic Infection

Jin et al.

Marek’s disease in chickens is caused by an oncogenic lymphotropic alphaherpesvirus, but is successfully controlled by vaccination using closely-related attenuated virus strains, such as the CVI988 vaccine strain.  Both CVI988 and virulent MDV strains  (such as RB1B) infect B and T lymphocytes in the first week of infection, and the reasons why CVI988 is non-pathogenic and induces a protective immune response, while virulent MDV strains cause lymphoid organ atrophy, immunosuppression, and ultimately tumour development, are not fully understood.

The authors addressed this question by collecting spleen and bursa from chickens experimentally infected with CVI988 or RB1B, and using qRT-PCR and RNA-seq to examine differences in host gene expression in the first week following infection.  Their results showed significant differences in expression of certain chemokines and cytokines between the spleens of CVI988-infected chickens and RB1B-infected chickens at 5 dpi, indicating that CVI988 induces stronger and earlier adaptive and innate immune responses than RB1B in the lymphoid organs during cytolytic infection.  Many of these differentially expressed genes between CVI988-infected and RB1B-infected birds had not previously been identified.  Therefore, this is an important and interesting study that provides novel information.

I have no direct experience of producing or analysing RNA-seq data, so I cannot comment in detail on the RNA-seq experiments.  However, overall, the studies seem to have been conducted thoroughly and using appropriate controls.  The Introduction gives an appropriate background to the study and is well-referenced.  The Materials & Methods section would benefit from some additional detail for clarification (see below).  The Results are, mostly (see below), clearly presented and appropriately analysed and interpreted, with clear figures.  The Discussion puts the results in context and is appropriately referenced.

The manuscript requires some editing to improve the spelling and use of the English language.  I have made some corrections and suggested changes in the accompanying edited PDF file.

Below are some points to address:

The words ‘undermined’ and ‘undermine’ are used several times in the manuscript, but it does not make sense. I think the words should be ‘examined’ (or ‘examine’) or ‘determined’ (or ‘determine’). The authors often refer to ‘CVI988-infected lymphocytes’ and ‘RB1B-infected lymphocytes’, but it would be better to say ‘lymphocytes from CVI988-infected chickens’ (because not every lymphocyte in an infected chicken will be infected). I have made these changes in the manuscript. The Introduction should make it clear that MDV vaccines are live (but non-pathogenic) viruses which replicate in the vaccinated chicken and persist for life of the chicken. Experimental animals and design: The distribution of chickens in the different groups is not clear. There were 105 chickens divided into three treatment groups.  35 were mock-treated controls.  Presumably 35 were infected with CVI988, and 35 were infected with RB1B, but this is not clear.  It is also stated that there were 10 birds per cage.  How were the birds grouped?  And five birds were culled on each of 7 days for collection of lymphoid organs.  Presumably this was five birds from each group (control, CVI988, RB1B) but this needs to be made clear. Separation of lymphocytes: How was a single cell suspension prepared from spleen and bursa before centrifugation over Histopaque? It should be made clear that the isolated lymphocytes were used for 3 purposes: extraction of DNA (for quantification of virus by real-time PCR), extraction of RNA using TRIzol (for high throughput sequencing), and extraction of RNA using RNeasy kit (for qRT-PCR to measure differentially-expressed genes). Absolute quantification of MDV in lymphocytes (line 123, and Figure 1): Absolute quantification requires normalisation of virus genome copies per certain number of lymphocytes, or per mass of tissue. It is not clear how the authors performed absolute quantification.  Was DNA extracted from a standard number of lymphocytes?  No internal reference gene seems to have been used in the qPCR. CVI988 and RB1B have some differences in the meq gene. Were the meq gene primers designed to amplify a conserved region that is common to both CVI988 and RB1B? Please provide a reference for the CVI988-BAC. Line 168: Please provide a reference for ‘Benjamini and Hochberg’, if possible. It should be made clear that (because virus replication peaked at 5 dpi for both viruses, and was highest in the spleen), spleen samples taken at 5 dpi were selected for detailed study by RT-qPCR and RNA-seq. Line 173: The abbreviation ‘dpi’ does not seem to have been defined anywhere. Please write it in full the first time it is mentioned. Lines 178 – 181: Were the differences in virus replication between spleen and bursa, and between CVI988 and RB1B, significant? The GO and KEGG analyses are not explained in ‘Materials & Methods’. What software was used? Figure 1: Do the values represent the mean from 5 birds? What are the bars? Standard deviation?  Standard error? Figure 3 (A, B & C): What does * indicate? Please mention this in the figure legend. Figure 3 (E & F): What is the q value? Please mention this in the figure legend. Lines 330-331: It is stated that “…the degree of host immune response is positively correlated with the viral load of MDV.” However, in this study, replication of RB1B was higher than CVI988, while CVI988 mounted a better immune response. Therefore, this is a negative correlation.

I have made some corrections to spelling and English language on a PDF file of the manuscript. Please see the attached.

Author Response

Cover letter
Ref: Viruses-701102
Title: Transcriptional Profiles Associated with Marek's Disease Virus in Bursa and Spleen Lymphocytes Reveal Contrasting Immune Responses during Early Cytolytic Infection
Journal: Viruses
Esteemed reviewer,
I am writing to you regarding the revised version of our manuscript submitted to the journal Viruses(viruses-701102). We highly appreciate you for yours time and constructive comments, which are greatly helpful for us to improve our manuscript. We have carefully revised our manuscript following the comments. Item-by-item responses to all the comments is uploaded with this revised manuscript. We would greatly appreciate your consideration for accepting our manuscript to publish in Viruses.Looking forward to your decision.

Sincerely yours

Yongqing Li

Yongqing Li
Ph.D., Professor
Email: [email protected]
Research Center for Infectious Disease in Livestock and Poultry, Beijing Academy of Agricultural and Forestry Sciences
Beijing 100197, People’s Republic of China

Point 1: The manuscript requires some editing to improve the spelling and use of the English language. I have made some corrections and suggested changes in the accompanying edited PDF file.
Response 1: We are thankful to the reviewer for his precious comments. According to the reviewer’s instructions, we have corrected the spelling mistakes and thoroughly revised the sentences in this manuscript.
Point 2: The words ‘undermined’ and ‘undermine’ are used several times in the manuscript, but it does not make sense. I think the words should be ‘examined’ (or ‘examine’) or ‘determined’ (or ‘determine’).
Response 2: We completely agree with the reviewer’s suggestions. According to the reviewer’s instructions, we have substituted the words ‘undermined’ and ‘undermine’ to ‘examined’ (or ‘examine’) or ‘determined’ (or ‘determine’).
Point 3: The authors often refer to ‘CVI988-infected lymphocytes’ and ‘RB1B-infected lymphocytes’, but it would be better to say ‘lymphocytes from CVI988-infected chickens’ (because not every lymphocyte in an infected chicken will be infected). I have made these changes in the manuscript.
Response 3: We are indebted to the reviewer’s comments. As per his comments, we have revised ‘CVI988-infected lymphocytes’ and ‘vvRB1B-infected lymphocytes’ to ‘lymphocytes from CVI988-infected chickens’ and ‘lymphocytes from vvRB1B-infected chickens’.
Point 4: The Introduction should make it clear that MDV vaccines are live (but non-pathogenic) viruses which replicate in the vaccinated chicken and persist for life of the chicken.
Response 4: We are highly grateful to the reviewer’s suggestions. In agreement to the reviewer’s comments, we have ncorported a new sentence stating ‘Especially, CVI988/Rispens (a naturally-attenuated vMDV strain that it establish persistent infection and life long immunity, effectively protecting against tumors and mortality) is known to be the most effective vaccine virus strain.’ In the introduction on line 88-90.
Point 5: Experimental animals and design: The distribution of chickens in the different groups is not clear. There were 105 chickens divided into three treatment groups.35 were mock-treated controls. Presumably, 35 were infected with CVI988, and 35 were infected with RB1B, but this is not clear. It is also stated that there were 10 birds per cage. How were the birds grouped? And five birds were culled on each of 7 days forthe collection of lymphoid organs. Presumably, this was five birds from each group (control, CVI988, RB1B) but this needs to be made clear.
Response 5: We agree with the reviewer’s comments. For the sake of clarity, we have restructured the distribution of chicks as following:‘One hundred and five SPF chicks (mixed-sex and one-day-old) were divided randomly into three groups (35 chickens in each group) and each group reared in separate cages. Chicks were euthanized humanely and samples from bursa and spleen were collected from five birds of each group (control,CVI988, RB1B) at regular intervals (1, 2, 3, 4, 5, 6 and 7 dpi).
Point 6: Separation of lymphocytes: How was a single cell suspension prepared from spleen and bursa before centrifugation over Histopaque? It should be made clear that the isolated lymphocytes were used for 3 purposes: extraction of DNA (for quantification of virus by real-time PCR), extraction of RNA using TRIzol (for high throughput sequencing), and extraction of RNA using RNeasy kit (for qRT-PCR to measure differentially-expressed genes).
Response 6: Many thanks for your comprehensive suggestion. The single-cell suspension of spleen and bursa were prepared as follows: ‘Spleen or bursa was macerated with a syringe plunger through a screen sieve to obtain a single cell suspension in PBS. pooled cells were placed in 50-ml tubes and clumps allowed to settle out for 5 min. The upper suspension was overlayed onto Histopaque 1077 density gradient media and centrifuged at 672g for 20 min at room temperature.’ The previous articles were referred for these experimental procedures, and we have incorporated the reference[52] in the revised manuscript. We have added the sentences‘the total DNA that was used for quantification of virus by real-time PCR’,‘total RNA that was used for high throughput sequencing, was extracted from lymphocytes using TRIzol Reagent’, ‘Total RNA that was used for detecting DEGs, was extracted from the lymphocytes of spleen and bursa using the RNeasy Mini Kit’in ‘Materials & Methods 2.3’,‘Materials & Methods 2.5’ and ‘Materials & Methods 2.7’, respectively.
Point 7: Absolute quantification of MDV in lymphocytes (line 123, and Figure 1): Absolute quantification requires normalisation of virus genome copies per certain number of lymphocytes, or per mass of tissue. It is not clear how the authors performed absolute quantification.Was DNA extracted from a standard number of lymphocytes? No internal reference gene seems to have been used in the qPCR.
Response 7: Many thanks for this comment. The genomic DNA was extracted from a population of 106 lymphocytes, and we have mentioned this in line 126 of the manuscript. In this study, the standard concentration of the DNA sample (50 ng/μL, line 128) was used to ensure the consistency of sample loading in the qPCR.
Point 8: CVI988 and RB1B have some differences in the meq gene. Were the meq gene primers designed to amplify a conserved region that is common to both CVI988 and RB1B? Please provide a reference for the CVI988-BAC.
Response 8: We appreciate this comment raised by the worthy reviewer. In this study, the meq gene primers were designed against a conserved region of both CVI988 and vvRB1B genome. The reference[53] was cited for the CVI988-BAC.
Point 9: Line 168: Please provide a reference for ‘Benjamini and Hochberg’, if possible. It should be made clear that (because virus replication peaked at 5 dpi for both viruses, and was highest in the spleen), spleen samples taken at 5 dpi were selected for detailed study by RT-qPCR and RNA-seq.
Response 9: We highly acknowledge the reviewer’s comments. We have added the reference[54] in ‘Materials & Methods 2.8’. We would like to point out that both spleen samples and bursal samples were taken at 5 dpi were selected for RNA-seq study, and the focus of this study is to analyze the RNA-seq result of spleen samples.
Point 10: Line 173: The abbreviation ‘dpi’ does not seem to have been defined anywhere. Please write it in full the first time it is mentioned.
Response 10: We recognize the reviewer’s comments. The full form of ‘dpi’ days post-infection is mentioned in line number 62.
Point 11: Lines 178 – 181: Were the differences in virus replication between spleen and bursa, and between CVI988 and RB1B, significant?
Response 11: We are grateful for raising this point. The differences in virus replication between spleen and bursa, and between CVI988 and RB1B were significant.The results of the difference analysis have been documented in Figure1.
Point 12: The GO and KEGG analyses are not explained in ‘Materials & Methods’. What software was used?
Response 12: Many thanks for this comment. We have mentioned the GO and KEGG analyses in‘Materials & Methods 2.6’; Goseq and KOBAS software were used for the analyses of GO annotationand KEGG pathway annotation, respectively.
Point 13: Figure 1: Do the values represent the mean from 5 birds? What are the bars? Standard deviation? Standard error?
Response 13: We thank the reviewer’s comments. The values represent the mean from 5 bird, and the data was showed as means ± SD of three independent experiments of per chiken and per indicated organ (*p < 0.05; **p < 0.01; ***p < 0.001; ns, no significant). The results of the difference analysis have been documented on the picture in Figure 1.
Point 14: Figure 3 (A, B & C): What does * indicate? Please mention this in the figure legend.
Response 14: We thank the reviewer’s comments.‘*’in Figure 3(A, B & C) indicates significant enrichment of GO term. As suggested by the reviewer, we have also added this in the figure legend.
Point 15: Figure 3 (E & F): What is the q value? Please mention this in the figure legend.
Response 15: We are highly grateful to the reviewer. In Figure 3 (E & F), q value (range 0-1) is the p-value that was corrected by multiple hypothesis tests. The closer the q value is to zero, the more significant the enrichment of KEGG pathway is. As suggested by the reviewer, we have also incorporated this in the figure legend.
Point 16: Lines 330-331: It is stated that “…the degree of host immune response is positively correlated with the viral load of MDV.” However, in this study, replication of RB1B was higher than CVI988, while CVI988 mounted a better immune response. Therefore, this is a negative correlation.
Response 16: We strongly agree with the reviewer’s suggestions. We have substituted the word ‘positively’ to ‘negatively’.
Point 17: I have made some corrections to spelling and English language on a PDF file of the manuscript. Please see the attached.
Response 17: We pay special thanks to the reviewer for your comprehensive efforts for this manuscript. Following your kind instructions, we have corrected the spellings and revised the sentences in this manuscript.

Reviewer 3 Report

The manuscript [Virus-701102], entitled “Transcriptional Profiles associated with Marek's disease Virus in Chicken B Cells Reveal Contrasting Immune responses during Early Cytolytic Infection” by Jin et al., reports finding on Marek’s disease virus (MDV)-induced transcriptomic profiles of the B cells and significant differences in expression of cytokine and chemokine genes in response to vaccine strain (CVI988) and non-vaccine strain (vvRB1B) MDV challenge.

This is an interesting study that was laid out to profile transcriptomic differences in lymphoid organs of bursa and spleen of chickens inoculated with CVI988/Rispens and RB1B MDV. The analyses of the data and the validation of the expressional data were done reasonably well. The content of the results reporting in this manuscript could be further improved.

Some specific concerns are listed below for consideration in revision.

      Major concerns:

Lack of statistics in statements of DEGs between treatment groups. The numeric expression values that appeared differing between treatment groups do not automatically support such claims of differences with statistical significance. Use of a threshold P value of 0.25 is questionable (line 234). What have been concluded in the end of introduction and discussion may not be fully supported by the data presented.

Minor concerns:

Minor grammatic errors are urged to be checked and corrected. For instance, “The MDV infection initiates by …” on line 52; “The second phase is termed as the latent that develops 7-9 dpi …” on line 59; “… using the deep sequencing approach …” on line 87; “… there were 408 up-regulated and 226 down-regulated DEGs were identified …” in lines 190-191. It is claimed that both viral copy number differed (higher) between the two and within each of primary lymphoid organs in lines 178-181 (showed in Fig. 1A-1D), but missing statistics that support the claims. Lines 186-88: Fig.S2 and Table S1 contents are inconsistent to the statement here. Suggested to spell out an abbreviation before using it (PPI on line 260).

Author Response

Cover letter
Ref: Viruses-701102
Title: Transcriptional Profiles Associated with Marek's Disease Virus in Bursa and Spleen Lymphocytes Reveal Contrasting Immune Responses during Early Cytolytic Infection
Journal: Viruses
Esteemed reviewer,
I am writing to you regarding the revised version of our manuscript submitted to the journal Viruses(viruses-701102). We highly appreciate you for yours time and constructive comments, which are greatly helpful for us to improve our manuscript. We have carefully revised our manuscript following the comments. Item-by-item responses to all the comments is uploaded with this revised manuscript. We would greatly appreciate your consideration for accepting our manuscript to publish in Viruses. Looking forward to your decision.

Sincerely yours

Yongqing Li

Yongqing Li
Ph.D., Professor
Email: [email protected]
Research Center for Infectious Disease in Livestock and Poultry, Beijing Academy of Agricultural and Forestry Sciences
Beijing 100197, People’s Republic of China

Point 1: Lack of statistics in statements of DEGs between treatment groups. The numeric expression values that appeared differing between treatment groups do not automatically support such claims of differences with statistical significance. Use of a threshold P value of 0.25 is questionable (line 234). What have been concluded in the end of introduction and discussion may not be fully supported by the data presented.
Response 1: We are humble to receive the comments from the reviewer. We would like to point out that for analyzing the results of this study, we used a higher threshold(p< 0.05 and |log2fold change|> 1.5) to select the genes with extremely significant differences from a large number of DEGs (Figure 2 and Table S1), and the DEGs determined by RT-qPCR were consistent with those of RNA-seq results (Figure 4), confirming the reliability of our RNA-seq data. For the threshold of ‘corrected P-value < 0.25’, there are 12 pathway in CVI988 group (Cytokine-cytokine receptor interaction, Influenza A,Toll-like receptor signaling pathway, Intestinal immune network for IgA production, AGE-RAGE signaling pathway in diabetic complications, Salmonella infection, NOD-like receptor signaling pathway, RIG-I-like receptor signaling pathway, Cytosolic DNA-sensing pathway, Herpes simplex infection, Arginine biosynthesis and Alanine, aspartate and glutamate metabolism)and 10 pathway in vvRB1B group(Cytokine-cytokine receptor interaction, Influenza A,Intestinal immune network for IgA production, Toll-like receptor signaling pathway, AGE-RAGE signaling pathway in diabetic complications, FoxO signaling pathway, Salmonella infection, Herpes simplex infection, Arginine, and proline metabolism and TGF-beta signaling pathway). In line number 274-275, we point out the pathways including ‘cytokine-receptor interaction, NOD-like receptor, RIG-I-likereceptor signaling, and Toll-like receptor signaling pathway.’ In order to make the conclusion more convincing, we have moved the sentences from Introduction to the Discussion and modified the end of discussion in the word usage aspect.
Point 2: Minor grammatic errors are urged to be checked and corrected. For instance, “The MDV infection initiates by …” on line 52; “The second phase is termed as the latent that develops 7-9 dpi …” on line 59; “… using the deep sequencing approach …” on line 87; “… there were 408 up-regulated and 226 down-regulated DEGs were identified …” in lines 190-191.
Response 2: We are highly indebted to the reviewer’s comments. As per the kind suggestions by the reviewer, we have carefully corrected the grammatical errors in our manuscript. We have significantly altered the sentences ‘The MDV infection initiates by …’ to ‘The MDV infection initiated by …’, ‘The second phase is termed as the latent that develops 7-9 dpi …’ to ‘The second phase is termed the latent phase, and develops 7-10 dpi in T cells which become …’, ‘…utilizing the deep sequencing approach …’ to ‘…using the deep sequencing approach …’, ‘… there were 408 up-regulated and 226 down-regulated DEGs were identified …’ to ‘408 up-regulated and 226 down-regulated DEGs were identified …’.
Point 3: It is claimed that both viral copy number differed (higher) between the two and within each of primary lymphoid organs in lines 178-181 (showed in Fig. 1A-1D), but missing statistics that support the claims.
Response 3: We thank the reviewer’s comments.The results of the difference analysis have been documented on the picture of Figure 1.
Point 4: Lines 186-88: Fig.S2 and Table S1 contents are inconsistent to the statement here.
Response 4: We agree with the reviewer’s comments. We have presented the DGEs of splenic lymphocytes in Fig.S2 and Table S1.
Point 5: Suggested to spell out an abbreviation before using it (PPI on line 260).
Response 5: As suggested by the reviewer, we have exchange ‘PPI’ to ‘protein-protein interaction (PPI)’.

Reviewer 4 Report

In the manuscript entitled “Transcriptional Profiles associated with Marek's disease Virus in Chicken B Cells Reveal Contrasting Immune responses during Early Cytolytic Infection” the authors provide comparative data on MDV mediated immune response in the lymphoid organs of chicken during the early phase of infection. Generally, the paper is well presented and argued, the conclusions drawn by the authors based on their findings are logical.

Comments, questions:

Line 50: Herpesviridae should be italicized.

Line 51 and later in the whole text: in vivo and in vitro should be italicized.

Line 89-94: These sentences should be moved from the Introduction to the Discussion.

Line 164: One-way or two-way ANOVA were used? What kind of post hoc test was applied?

Much more DEGs were observed in lymphocytes infected with the vaccine strain. Comparing the genomes of the two MDV strains, what could be the putative reasons of this significant difference in the number of the DEGs?

Editing of English language and style is required.

Author Response

Cover letter
Ref: Viruses-701102
Title: Transcriptional Profiles Associated with Marek's Disease Virus in Bursa and Spleen Lymphocytes Reveal Contrasting Immune Responses during Early Cytolytic Infection
Journal: Viruses
Esteemed reviewer,
I am writing to you regarding the revised version of our manuscript submitted to the journal Viruses(viruses-701102). We highly appreciate you for yours time and constructive comments, which are greatly helpful for us to improve our manuscript. We have carefully revised our manuscript following the comments. Item-by-item responses to all the comments is uploaded with this revised manuscript. We would greatly appreciate your consideration for accepting our manuscript to publish in Viruses. Looking forward to your decision.

Sincerely yours

Yongqing Li

Yongqing Li
Ph.D., Professor
Email: [email protected]
Research Center for Infectious Disease in Livestock and Poultry, Beijing Academy of Agricultural and Forestry Sciences
Beijing 100197, People’s Republic of China

Point 1: Line 50: Herpesviridae should be italicized.
Line 51 and later in the whole text: in vivo and in vitro should be italicized.
Response 1: As suggested by the reviewer, we have italicized “Herpesviridae” and “in vivo and in vitro” in the whole text.
Point 2: Line 89-94: These sentences should be moved from the Introduction to the Discussion.
Response 2: We thank the reviewer’s comments. As suggested by the reviewer, we have moved the sentences from the Introduction to the end of Discussion.
Point 3: Line 164: One-way or two-way ANOVA were used? What kind of post hoc test was applied?
Response 3: We thank the reviewer’s comments. Statically ‘one-way ANOVA’ was employed, and the ‘Student-Newman-Keuls (S-N-K)’ was applied to the post hoc test in this study.
Point 4: Much more DEGs were observed in lymphocytes infected with the vaccine strain. Comparing the genomes of the two MDV strains, what could be the putative reasons of this significant difference in the number of the DEGs?
Response 4: Many thanks for critically addressing this point. According to the results in our study, the difference of virus replication in the early stage is one of the reasons that were presumed to have the greatest possibility to be associated with the significant difference in the number of the DEGs. And that, previously, pp38 and vIL-8 were identified as key factors involved in the virus replication at the early stage. Viral replication in B cells is defined as semi-productive lytic viral replication (2-7 days). Lytic activity due to viral replication has been linked to phosphoprotein 38 (pp38) activities. pp38 role as an early immediate gene is defined only in lymphocytes, specifically B cells and T cells. It has been shown that an MDV rMd5Deltapp38 deletion mutant for pp38 lacked the ability to induce cytolytic activity characterized by B cell apoptosis. This is in accordance with the notion that the recruitment of lymphocytes such as B cells to the site of viral replication is a key step for transmission of virus and dissemination. Moreover, vIL-8, one of the defined viral chemokine that is similar to CXCL13 and is involved in recruiting immune systems cells (B cells and T cells) to the site of viral replication. Deletion mutants of vIL-8 (RB1BvIL-8ΔsmGFP), when tested in vivo, showed a reduced capacity to successfully infect lymphocytes and induce lytic infection. A lack of vIL-8, therefore, result in impaired ability to recruit B cells and as well as an observable reduction in cytolytic activity due to reduced viral titer and dissemination by lymphocytes. Therefore, we speculate that some factors (pp38 and vIL-8, etc.) affecting virus replication are related to the significant difference in the number of DEGs.
Point 5: Editing of English language and style is required.
Response 5: Keeping in view the comments of the reviewer, we have edited and carefully revised the English language and style in our manuscript.

Round 2

Reviewer 1 Report

The authors are wrong in assuming that “spenic lymphocytes are mainly constituted of B-lymphocytes”, it has long been established that a diverse population of lymphocytes, including a variety of T-cells (e.g. CD4+, CD8+, Treg, etc.) populate the spleen (for Reference: Jeurissen SH. Structure and function of the chicken spleen. Res Immunol. 1991 May;142(4):352-355). It also cannot be assumed that by using a “higher threshold” for selecting DEG this will somehow identify B lymphocyte specific responses. It is also incorrect to speculate that “there is no significant difference for the genes with extremely significant differences between mixed lymphocytes and B lymphocytes” without any experimental evidence for support. There is also a diverse population of non-lymphoid immune cells in the spleen, which will also be present in the buffy coat from Histopaque preparations. Even though T-cells many not, to a large extent at least, be directly infected by MDV, in the earliest stages of infection, they and other immune cells with still respond to infection and will therefore significantly contribute to the transcriptome. Though the authors have made some revisions to the manuscript to acknowledge these limitations, the discussion is still too overreaching, and does not have enough experimental support for many of conclusions.

Author Response

Ref: Viruses-701102
Title: Transcriptional Profiles Associated with Marek's Disease Virus in Bursa and Spleen Lymphocytes Reveal Contrasting Immune Responses during Early Cytolytic Infection
Journal: Viruses

Esteemed reviewer,

I am writing to you regarding the revised and updated version of our manuscript submitted to the journal Viruses (viruses-701102). We highly appreciate you for your time and constructive comments, which definitely helped us to improve our manuscript. Now, we have carefully revised our manuscript following the comments. One by one response to all the comments is uploaded with this revised manuscript. We would greatly appreciate your consideration for accepting our manuscript to publish in Viruses. Looking forward to your decision.

Sincerely yours

Yongqing Li

Yongqing Li

Ph.D., Professor

Research Center for Infectious Disease in Livestock and Poultry, Beijing Academy of Agricultural and Forestry Sciences

Beijing 100197, People’s Republic of China

Point: The authors are wrong in assuming that “splenic lymphocytes are mainly constituted of B-lymphocytes”, it has long been established that a diverse population of lymphocytes, including a variety of T-cells (e.g. CD4+, CD8+, Treg, etc.) populates the spleen (for Reference: Jeurissen SH. Structure and function of the chicken spleen. Res Immunol. 1991 May;142(4):352-355). It also cannot be assumed that by using a “higher threshold” for selecting DEG this will somehow identify B lymphocyte-specific responses. It is also incorrect to speculate that “there is no significant difference for the genes with extremely significant differences between mixed lymphocytes and B lymphocytes” without any experimental evidence for support. There is also a diverse population of non-lymphoid immune cells in the spleen, which will also be present in the buffy coat from Histopaque preparations. Even though T-cells many not, to a large extent at least, be directly infected by MDV, in the earliest stages of infection, they and other immune cells with still respond to infection and will therefore significantly contribute to the transcriptome. Though the authors have made some revisions to the manuscript to acknowledge these limitations, the discussion is still too overreaching, and does not have enough experimental support for many of conclusions.

Response: We are humble and really grateful for the reviewer’s scientific, professional and precious comments. In agreement with the reviewer’s comments, we have restructured the discussions of our manuscript in a precise order that the results can support them. The details of the amendments are as follows:

Lines 21-25: We have re-written the sentence as ‘B cells are the primary target of MDV replication during the early cytolytic phase of infection. The bursa of Fabricius is the primary site in which the maturation and differentiation of B lymphocytes take place, and the spleen is the major location of B cell immune responses. So far, limited information is available on MDV mediated response in B cells during the early cytolytic phase of infection’ to ‘Due to lymphocytes of bursa of Fabricius and spleen are prime targets of MDV replication during the early cytolytic phase of infection, the immune response in bursa and spleen should be foundation of late immunity induced by MDV. However, the mechanism of how MDV-mediated host immune response in lymphocytes on early stage is poorly understand.

Line 52: We have changed the word ‘B lymphocytes’ to ‘ Lymphocytes’

Lines 96-99: We have changed the sentence ‘However, the vaccine-mediated immune response is still poorly studied, especially how these vaccines activate the B cell repertoires during the cytolytic phase of infection.’ to ‘However, the vaccine-mediated immune response, and in particular, how these vaccines activate the host immune response during the cytolytic phase of infection, is still poorly studied.’

Line 378: We have changed the sentence ‘the majority of which are B lymphocytes’ to ‘lymphocytes in spleen/bursa’

Lines 406-408 and 414-416: We have combined the two sentences ‘Our findings further validated that MDV infection significantly increased IL-4 production leading to the activation of mature B cells in the bursa.’ and ‘The upregulation of these cytokines in the present study further leads to the activation of B cell to elicit a broad antiviral environment within the host cells.’ into ‘The upregulation of these cytokines in the present study may further modulate the B and T lymphocytes to elicit a broad antiviral environment within the host cells.’

Lines 424-426: We have altered the sentence ‘The up-regulation of IL-10, IFN-γ, and STAT1 in this article suggested that MDV infection regulates B cell functions in vivo.’ to ‘The up-regulation of IL-10, IFN-γ, and STAT1 in this article suggest that MDV infection should regulate B lymphocytes functions in vivo.’

Reviewer 3 Report

The revision was done pretty thoroughly with significant improvement. There are minor punctuation/space issues to be further corrected.

No other major specifics to be addressed. 

Author Response

Ref: Viruses-701102
Title: Transcriptional Profiles Associated with Marek's Disease Virus in Bursa and Spleen Lymphocytes Reveal Contrasting Immune Responses during Early Cytolytic Infection
Journal: Viruses

Esteemed reviewer,

I am writing to you regarding the revised and updated version of our manuscript submitted to the journal Viruses (viruses-701102). We highly appreciate you for your time and constructive comments, which definitely helped us to improve our manuscript. We have acquired the professional English Language Editing Services of MDPI (English Editing Certificate 16812). Now, we have carefully revised our manuscript following the comments. One by one response to all the comments is uploaded with this revised manuscript. We would greatly appreciate your consideration for accepting our manuscript to publish in Viruses. Looking forward to your decision.

Sincerely yours

Yongqing Li

Yongqing Li

Ph.D., Professor

Research Center for Infectious Disease in Livestock and Poultry, Beijing Academy of Agricultural and Forestry Sciences

Beijing 100197, People’s Republic of China

Point: The revision was done pretty thoroughly with significant improvement. There are minor punctuation/space issues to be further corrected.

Response: We are highly grateful to the reviewer’s suggestions. In order to improve the language, we have acquired the professional editing by MDPI's English editing services for this manuscript. We are hoping that now it will be accepted for publication in Viruses.